# Sensitivity Analysis of Wing Geometric and Kinematic Parameters for the Aerodynamic Performance of Hovering Flapping Wing

**Xinyu Lang** [1,2,3] 🆔, **Bifeng Song** [1,2,3], **Wenqing Yang** [1,2,3] 🆔, **Xiaojun Yang** [1,2,3,*] 🆔 and **Dong Xue** [1,2,3]

1   National Key Laboratory of Science and Technology on Aerodynamic Design and Research, School of Aeronautics, Northwestern Polytechnical University, Xi'an 710072, China
2   Research & Development Institute of Northwestern Polytechnical University in Shenzhen, Shenzhen 518057, China
3   Yangtze River Delta Research Institute of Northwestern Polytechnical University, Taicang 215400, China
*   Correspondence: xjyang518@nwpu.edu.cn

**Abstract:** The wing planform and flapping kinematics are critical for the hovering flight of flapping wing micro air vehicles (FWMAVs). The degree of influence of wing geometry and kinematic parameters on aerodynamic performance still lacks in-depth analysis. In this study, a sensitivity analysis was conducted based on the quasi-steady aerodynamic model. Each parameter was investigated independently by using the control variable method. The degree of each variable's influence on lift, power, and power loading is evaluated and compared. Furthermore, detailed exponential relationships were established between the parameters and the corresponding aerodynamic properties. It is found that, for the geometric parameters, wing area has the greatest influence on lift, and the distribution of area has the most visible effect on aerodynamic power. All geometric parameters are negatively correlated with power loading. For the kinematic parameters, flapping frequency, compared with sweeping amplitude, results in faster lift growth and slower drop in power loading, while their influence on aerodynamic power is nearly comparable. A moderate pitching amplitude with advanced rotation will maximize the lift. For the flapping trajectory, lift and power loading are primarily affected by the shape of the pitching motion rather than the sweeping motion. But the sweeping motion seems to dominate the power consumption. The research in this paper is helpful to understand the effect of each parameter and provide theoretical guidance for the development of FWMAVs.

**Keywords:** flapping wing; hovering flight; sensitivity analysis; geometric parameters; kinematic parameters

## 1. Introduction

Flapping wing micro air vehicles (FWMAVs), especially those with hovering flight capability, have attracted growing interest in the past few years [1,2]. Various insect-inspired or hummingbird-like FWMAVs have been developed, such as Nano Hummingbird [3], KUBeetle [4], Colibri robot [5], etc., for sustainable hovering flight and potential applications. The improvement of the hovering FWMAVs with higher lift force and lower power consumption is the persistent pursuit of researchers. Numerous numerical and experimental efforts have been made to design and optimize the flapping wings which are the primary component of FWMAVs to generate aerodynamic force [6–9]. It has been demonstrated that the wing planform and the flapping kinematics function as the dominant factors in the aerodynamic analysis of hovering flight [10–14].

The shape of the flapping wing planform always determines the aerodynamic force and efficiency. Uniform or dimensionless parameters, such as wing area $S$, aspect ratio $AR$ and dimensionless radius of the second moment of area $r_2$ are often used to describe or define the wing geometry. Effect of these parameters on the aerodynamic performance

of flapping wing has been explored widely. Researches on the aspect ratio $AR$ present contradictory results. Some studies indicate that the increase of $AR$ will lead to the monotonic variation of lift force [11,15,16], while others present an optimum value around $AR = 4$ [17,18]. This discrepancy might result from the flexibility of the wing. Meanwhile, the flow structures around the wing appear to be sensitive to $AR$, which will in turn affect the aerodynamic force. Additionally, the wing area also presents an apparent influence on the aerodynamic performance. The experiment conducted by Nan et al. [9] and Deng et al. [19] has shown that a larger wing area will create more lift, as well as power consumption. The dimensionless radius of the second moment of area $r_2$, which reflects the distribution of wing area or chord length, is another crucial parameter for wing shape. Research by Shahzad et al. [11] and Broadley et al. [13] indicates that the wing with a greater outboard area can produce higher lift at the cost of more power. It can be found that all the above geometric parameters are able to affect the aerodynamic performance of flapping wing. However, few studies have compared the degree of influence of each parameter.

The flapping kinematics also present close relationship with the hovering lift and maneuverability. The feature parameters include flapping frequency, sweeping amplitude, pitching trajectory, etc. The flapping frequency shows an allometric positive correlation with lift and power as summarized by Nan et al. [9] and Nguyen et al. [20]. According to Phan et al. [21] who evaluated how the sweeping amplitude affects the flight efficiency of beetles, a larger sweeping amplitude is more beneficial for power requirement of the same vertical force production than a smaller amplitude. The introduction of pitching motion, whether passive rotation or active morphing, shows great potential advantages in lift and efficiency. Result of Lua et al. [22] and Sane and Dickinson [23] showed that an advance in pitching motion with a moderate angle of attack is the high-lift behavior. Recently, the impact of flapping trajectories is gradually attracting attention. A triangular sweeping motion accompanied by a trapezoidal pitching motion, which is similar to the real insect wing kinematics, will be beneficial for aerodynamic efficiency [14,24]. Although all of these kinematic parameters have a non-negligible influence on aerodynamic performance, few studies have provided a comprehensive answer to which parameter is more important for hovering flight.

For the aerodynamic design and optimization of FWMAVs with various parameters, it is necessary to evaluate the degree of influence of each variable. Subsequently, the most crucial factors can be identified to speed up the design process. Nevertheless, few researchers have focused on the comparison of the influence degree. The limitation might be that computational fluid dynamics and experimental methods are still very difficult to deal with a large number of cases with multiple design variables. Nowadays, the quasi-steady aerodynamic model offers a new choice, which can greatly improve design efficiency while ensuring reliability [25–27]. Moreover, it has been successfully applied to the design and optimization of the hovering FWMAVs [28–30].

To explore the degree of influence of geometric and kinematic parameters on the hovering performance of FWMAVs, a sensitivity analysis was performed based on the quasi-steady aerodynamic model. Using the control variable method, various cases were carefully designed to analyze each parameter independently. The degree of each variable's influence on lift, power, and power loading is evaluated and compared. Furthermore, the function of variables and corresponding aerodynamic properties are established. The present research will help to identify the variables with greater influence and offer theoretical guidance for the development of FWMAVs.

## 2. Materials and Methods

### 2.1. Key Parameters Selection

According to the blade element theoretical analysis by Ellington [10] and the experimental research of Lee et al. [6], the cycle-averaged force, power, and power loading can be written as follows:

$$
\begin{aligned}
L &= \tfrac{1}{2}\rho U^2 S C_L = \tfrac{1}{2}\rho \int_0^R (4\pi r f \psi_m)^2 C_L c\,\mathrm{d}r \\
&= 8\rho\pi^2 f^2 \psi_m^2 \int_0^R c r^2 \mathrm{d}r \overline{C}_L = 8\rho\pi^2 f^2 \psi_m^2 r_2^2 R^2 S\overline{C}_L \quad, \\
&= 8\rho\pi^2 f^2 \psi_m^2 r_2^2 A R S^2 \overline{C}_L
\end{aligned}
\tag{1}
$$

$$
\begin{aligned}
P &= \tfrac{1}{2}\rho U^2 S C_D U = \tfrac{1}{2}\rho \int_0^R (4\pi r f \psi_m)^3 C_D c\,\mathrm{d}r \\
&= 32\rho\pi^3 f^3 \psi_m^3 r_3^3 S R^3 \overline{C}_D = 32\rho\pi^3 f^3 \psi_m^3 r_3^3 A R S^2 R\overline{C}_D \quad, \\
&= 32\rho\pi^3 f^3 \psi_m^3 r_3^3 A R^{\frac{3}{2}} S^{\frac{5}{2}} \overline{C}_D
\end{aligned}
\tag{2}
$$

$$
\begin{aligned}
PL &= \frac{L}{P} = \frac{8\rho\pi^2 f^2 \psi_m^2 r_2^2 A R S^2 \overline{C}_L}{32\rho\pi^3 f^3 \psi_m^3 r_3^3 A R^{\frac{3}{2}} S^{\frac{5}{2}} \overline{C}_D} = \frac{r_2^2 \overline{C}_L}{4\pi f \psi_m r_3^3 R \overline{C}_D} \\
&= \frac{r_2^2 \overline{C}_L}{4\pi f \psi_m r_3^3 A R^{\frac{1}{2}} S^{\frac{1}{2}} \overline{C}_D}
\end{aligned}
\tag{3}
$$

where $f$ is the flapping frequency, $\psi_m$ is the sweeping amplitude, $AR$ is aspect ratio, $S$ is wing area, $\overline{C}_L$ and $\overline{C}_D$ is the cycle-averaged lift and drag force coefficient, $r_2$ and $r_3$ is the dimensionless radius of the second and third moment of wing area. It can be found that the parameters related to aerodynamic performance of hovering flapping wing can be grouped into geometric parameters and kinematic parameters. The geometric parameters include $AR$, $S$, $r_2$, and $r_3$. The kinematic parameters incorporate $f$ and $\psi_m$.

The above equations present an estimated relationship between the aerodynamic performance and sensitivity parameters and offer initial guidance in the subsequent study. However, these equations only considered the translation term from the quasi-steady point of view, while the rotation and additional mass components that may play an important role were neglected. Consequently, this estimated relationship between parameters and aerodynamic performance might not be accurate enough. Meanwhile, due to the introduction of additional variables such as $r_3$, the influence of some parameters is not intuitive. Additionally, the existence of force coefficients $\overline{C}_L$ and $\overline{C}_D$, which might vary with the wing shape or flight state [21,26], will lead to inaccurate results if the above equations are directly used for sensitivity analysis. Therefore, it is still necessary to carefully explore the relationship between sensitivity variables and aerodynamic performance.

### 2.2. Wing Geometric Parameters

Wing planform and geometric parameters adopted in this study were presented in Figure 1. The rectangular configuration is selected as the benchmark geometry because the rectangular shape can be easily parameterized to study the influence of geometric parameters.

The wing is assumed to be a thin rigid plate combined with sweeping and pitching motions. The sweeping axis is parallel to the wing root, and the wing pitches around the leading edge. As presented in Figure 1, the total length from the sweeping axis to the wing tip is defined as $R$, the wing span is $B$, and the offset from the wing root to the pitching axis is denoted as $\Delta R = R - B$. The chord length at the wing root and wing tip is defined as $c_R$ and $c_T$ respectively, thus the mean chord is given by $\overline{c} = (c_R + c_T)/2$.

According to the quasi-steady blade element model proposed by Ellington [10], wing area $S$, aspect ratio $AR$, and dimensionless radius of the second moment of area $r_2$ are selected as the key geometric parameters in this study. Since $r_3$ and $r_2$ can both reflect the distribution of the wing area, $r_3$ is not considered in this study. Therefore, these three key parameters are established for the geometric parameters sensitivity analysis. Additionally,

taper ratio ($\lambda = c_R/c_T$) is also considered under the research framework of $r_2$. The wing area $S$ can be calculated by integrating the strip of wing area and is given by:

$$S = \int_{\Delta R}^{R} c \, dr, \tag{4}$$

Thus, the aspect ratio $AR$ can be expressed as:

$$AR = \frac{R^2}{S}, \tag{5}$$

The dimensionless radius of the second moment of area $r_2$ is a crucial parameter that denotes the distribution of the chord along the wing or wing area and is defined by:

$$r_2 = \frac{R_2}{R} = \frac{\sqrt{\frac{1}{S} \int_0^R c r^2 \, dr}}{R}, \tag{6}$$

where $R_2$ is the radius of the second moment of area.

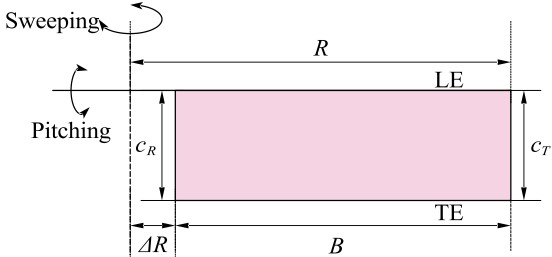

**Figure 1.** Schematic diagram of the wing planform and geometric parameters. LE means leading edge, and TE means trailing edge.

By referring to the derivation process of the geometric parameters aforementioned, it can be found that there is a potential relationship between each parameter. That is to say, changing one variable may cause changes in others, so the variables being investigated need to be carefully designed to meet the control variates principle. Therefore, when designing geometric parameters, only one variable is changed at a time.

The designed wings with various $AR$ and $S$ are listed in Table 1. In Group 1, the wing shape remained rectangular when the $AR$ is changed. The wing area $S$ remains constant by changing the wing span and chord length synchronously. In Group 2, the variable is the wing area $S$, and the aspect ratio $AR$ is kept constant by synchronously increasing the span and chord length. It should be noted that to isolate the influence of wing shape change to $r_2$, the wing offset from the root $\Delta R$ is assumed to be zero.

**Table 1.** List of designed wings with various $AR$ and $S$.

| | Wing Name | Wing Tip Radius $R$ (mm) | Aspect Ratio $AR$ | Taper Ratio $\lambda$ | $r_2$ | Wing Area $S$ (mm²) |
|---|---|---|---|---|---|---|
| Group 1 | 1 | 81.6 | 2 | | | |
| | 2 | 91.3 | 2.5 | | | |
| | 3 | 100.0 | 3 | | | |
| | 4 | 108.0 | 3.5 | | | |
| | 5 | 115.5 | 4 | 1 | 0.577 | 3333 |
| | 6 | 122.5 | 4.5 | | | |
| | 7 | 129.1 | 5 | | | |
| | 8 | 135.4 | 5.5 | | | |
| | 9 | 141.4 | 6 | | | |

**Table 1.** *Cont.*

| | Wing Name | Wing Tip Radius $R$ (mm) | Aspect Ratio $AR$ | Taper Ratio $\lambda$ | $r_2$ | Wing Area $S$ (mm$^2$) |
|---|---|---|---|---|---|---|
| Group 2 | 10 | 81.6 | | | | 2220 |
| | 11 | 91.3 | | | | 2779 |
| | 12 | 100.0 | | | | 3333 |
| | 13 | 108.0 | | | | 3888 |
| | 14 | 115.5 | 3 | 1 | 0.577 | 4447 |
| | 15 | 122.5 | | | | 5002 |
| | 16 | 129.1 | | | | 5556 |
| | 17 | 135.4 | | | | 6111 |
| | 18 | 141.4 | | | | 6665 |

Another crucial parameter, $r_2$, determines the area distribution of the wing, which has been proved to play an important role in the lift generation [13]. By referring to Equation (6), one can find that $r_2$ is closely related to the local chord length of the wing. Therefore, $r_2$ was modulated by changing the local chord length distribution in this study.

First, maximum chord length $c_{max}$ and its dimensionless spanwise location $lc/R$ are considered to adjust $r_2$. Table 2 presents wings with various $r_2$ by changing $c_{max}$ and $lc/R$, while the taper ratio $\lambda$ is kept constant.

**Table 2.** List of wings with different $r_2$ by changing maximum chord length $c_{max}$ and its spanwise location.

| | Wing Name | Root Chord $c_R$ (mm) | Ratio of $c_{max}$ to $c_R$ $c_{max}/c_R$ | Spanwise Location of $c_{max}$ $lc/R$ | Taper Ratio $\lambda$ | $r_2$ |
|---|---|---|---|---|---|---|
| Group 3 | 19 | | | 0.125 | | 0.557 |
| | 20 | | | 0.25 | | 0.561 |
| | 21 | | | 0.375 | | 0.566 |
| | 22 | 27.8 | 1.4 | 0.5 | 1 | 0.573 |
| | 23 | | | 0.625 | | 0.579 |
| | 24 | | | 0.75 | | 0.586 |
| | 25 | | | 0.875 | | 0.594 |
| Group 4 | 26 | 33.3 | 1.0 | | | 0.577 |
| | 27 | 31.7 | 1.1 | | | 0.573 |
| | 28 | 30.3 | 1.2 | | | 0.569 |
| | 29 | 29.0 | 1.3 | | | 0.565 |
| | 30 | 27.8 | 1.4 | 0.25 | 1 | 0.561 |
| | 31 | 26.7 | 1.5 | | | 0.558 |
| | 32 | 25.6 | 1.6 | | | 0.555 |
| | 33 | 24.7 | 1.7 | | | 0.552 |
| | 34 | 23.8 | 1.8 | | | 0.549 |
| Group 5 | 35 | 33.3 | 1.0 | | | 0.577 |
| | 36 | 31.7 | 1.1 | | | 0.580 |
| | 37 | 30.3 | 1.2 | | | 0.582 |
| | 38 | 29.0 | 1.3 | | | 0.584 |
| | 39 | 27.8 | 1.4 | 0.75 | 1 | 0.586 |
| | 40 | 26.7 | 1.5 | | | 0.588 |
| | 41 | 25.6 | 1.6 | | | 0.590 |
| | 42 | 24.7 | 1.7 | | | 0.591 |
| | 43 | 23.8 | 1.8 | | | 0.593 |

Additionally, $r_2$ will also change with the variation of the taper ratio $\lambda$ and presented in Table 3, indicating that the rectangular wing transforms into a trapezoidal shape. Therefore, the influence of $r_2$ by modifying the taper ratio $\lambda$ has also been investigated. Noteworthy,

in all the cases of $r_2$ mentioned above, $R$ is set to 100 mm, $S$ is 3333 mm$^2$, and $\Delta R$ is set to zero.

**Table 3.** Wings with different $r_2$ by changing the taper ratio $\lambda$.

| | Wing Name | Root Chord $c_R$ (mm) | Taper Ratio $\lambda$ | $r_2$ |
|---|---|---|---|---|
| Group 6 | 44 | 41.7 | 0.6 | 0.540 |
| | 45 | 39.2 | 0.7 | 0.551 |
| | 46 | 37.0 | 0.8 | 0.561 |
| | 47 | 35.1 | 0.9 | 0.570 |
| | 48 | 33.3 | 1.0 | 0.577 |
| | 49 | 31.7 | 1.1 | 0.584 |
| | 50 | 30.3 | 1.2 | 0.590 |
| | 51 | 29.0 | 1.3 | 0.596 |
| | 52 | 27.8 | 1.4 | 0.601 |

*2.3. Wing Kinematic Parameters*

The setup of coordinate systems and the definition of wing motions are given in Figure 2. An inertial frame ($Oxyz$) and wing-fixed frame ($O_w x_w y_w z_w$) are introduced. The flapping movements of the flat plate are composed of the sweeping around $z$ and pitching motion around $y_w$. Two Euler angles, including the sweeping angle $\psi$ and pitching angle $\theta$, are introduced. As shown in Figure 2, the sweeping angle $\psi$ is defined as the position of the wing relative to the midstroke in the horizontal stroke plane. The pitching angle $\theta$ is the rotation angle of the wing relative to the horizontal stroke plane. The angle of attack $\alpha$ is defined as the angle between the sweeping direction and the chord line. According to the geometric relationship shown in Figure 2b, the angle of attack $\alpha$ can be expressed as:

$$\begin{cases} \alpha = \theta, \text{ downstroke} \\ \alpha = \pi - \theta, \text{ upstroke} \end{cases} \tag{7}$$

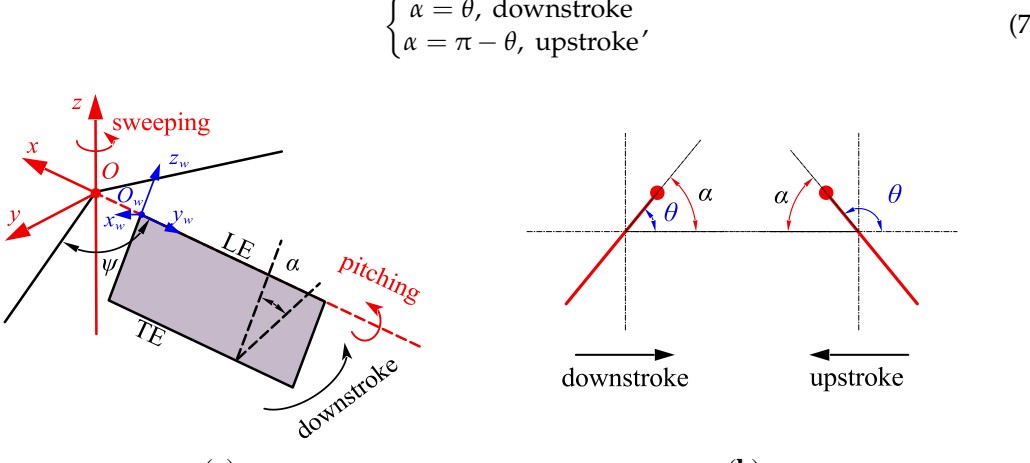

(**a**)          (**b**)

**Figure 2.** Definition of the flapping angle variables. (**a**) Schematic diagram of the sweeping and pitching motion and coordinate systems. (**b**) Definition of the angle of attack $\alpha$ and rotation angle $\theta$. The red dot means the leading edge. The black arrow line represents the flapping direction.

It should be noted that the elevating motion, defined as motion deviating from the stroke plane, can be observed in the experimental measurement of insect wings kinematics [31,32]. However, the amplitude of elevating motion is quite smaller than that of the other two motions. Meanwhile, a small elevating motion has been proven to have minimal aerodynamic effects compared with the other two motions [23,33]. Therefore, the elevating motion has been isolated and only the sweeping and pitching motion are incorporated.

The flapping kinematic proposed by Berman and Wang [34] is adopted in this study, which presents very similar wing trajectory as the measured kinematic from the hovering flapping wing [35]. The active sweeping and pitching kinematics are given as follows:

$$\psi(t) = \frac{\psi_m}{\sin^{-1}K}\sin^{-1}(K\cos(2\pi ft)),\tag{8}$$

$$\theta(t) = \frac{\theta_m}{\tanh C_\theta}\tanh(C_\theta\cos(2\pi ft - \varphi)) + \theta_0,\tag{9}$$

where $\theta_m$ is the pitching amplitude, $\varphi$ is the phase angle between sweeping and pitching motion, and $\theta_0$ is the offset of the mean pitching angle from the horizontal stroke plane. As shown in Figure 3, the other two parameters, $K$ and $C_\theta$, determine the shape of sweeping and pitching trajectory respectively.

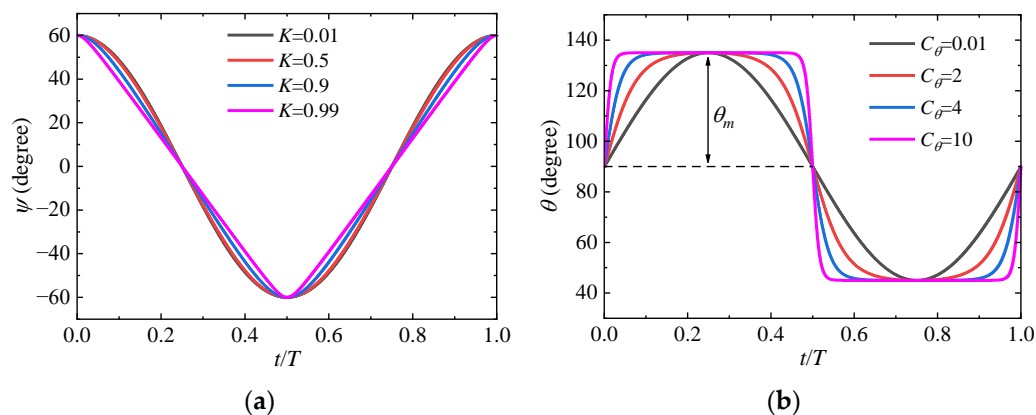

**Figure 3.** Flapping kinematics of different parameters. (**a**) Sweeping angle of different $K$. (**b**) Pitching angle of different $C_\theta$. $T$ is the flapping period.

The Reynolds number is given by:

$$Re = \frac{U_{ref}\bar{c}}{v},\tag{10}$$

where the reference velocity $U_{ref} = 4\pi f\psi_m R_2/180$, $v$ is the kinematic viscosity of air. As the flapping frequency in this study ranges from 10 Hz to 30 Hz, the Reynolds number varies from 10,000 to 30,000.

### 2.4. Sensitivity Analysis of Parameters

Sensitivity analysis can be performed to investigate the effects of various parameters on the flapping wing's aerodynamic performance. More crucially, to determine the extent to which each parameter influences aerodynamic performance. To explore the effect of wing shape and flapping motion on the hovering performance, the geometric and kinematic parameters are considered for the sensitivity analysis. While conducting the analysis process, only one parameter is changed and all other parameters are maintained constant.

The geometric parameters for the sensitivity analysis consist of $AR$, $S$, and $r_2$. The kinematic parameters for the sensitivity include $f$, $\phi_m$, $\theta_m$, $\varphi$, $K$, and $C_\theta$. The dimensionless form of sensitivity [36] can be expressed as:

$$S_{AF} = \frac{\partial y(x)/y(x)}{\partial x/x}.\tag{11}$$

During the sensitivity calculation, the variable $x$ changes by an increment $\Delta x$ each time, and the corresponding dependent variable $y$ changes by $\Delta y$. So, the sensitivity analysis can be calculated as:

$$S_{AF} = \frac{\Delta y(x)/y(x)}{\Delta x/x}. \tag{12}$$

It should be noted that when calculating the $S_{AF}$ in different frequency states, $y$ is considered as the value at that corresponding frequency, rather than a constant value at a fixed frequency. From the sensitivity analysis, it is helpful to identify the most important variable in lift generation and energy saving. Consequently, researchers can only focus on the predominant parameters to speed up the design process.

### 2.5. Aerodynamic Model, Force and Power Calculation

While conducting the sensitivity analysis, the aerodynamic forces and power consumption under various parameters and flapping frequency need to be calculated. Therefore, a quasi-steady aerodynamic model developed by Lee et al. [26] was selected to deal with the large amount of calculation work. Translational, rotational, and added mass load are the three terms that make up this aerodynamic model. The semi-empirical coefficients in the model were obtained from numerical simulations. This aerodynamic model has been validated by various numerical simulations and experimental measurements. As the model could yield reasonable force and power predictions over a wide range of wing geometric and flapping kinematic situations, it has been widely used in force calculation and optimization [29,37]. Thus, the aerodynamic model is suitable for solving the analysis problem considering various geometric and kinematic parameters.

As the elevating motion is not considered in this study, the quasi-steady aerodynamic model can be written as follows:

$$F_L = F_{L,tr} + (F_{rot,1} + F_{rot,2} + F_a)\cos\alpha, \tag{13}$$

$$F_D = F_{D,tr} + (F_{rot,1} + F_{rot,2} + F_a)\sin\alpha, \tag{14}$$

where $F_L$ and $F_D$ are the total lift and drag force, $F_{L,tr}$ and $F_{D,tr}$ are the translation lift and drag force respectively, $F_{rot,1}$ and $F_{rot,2}$ are the rotation force, and $F_a$ is the added mass force. Each force component can be expressed in detail as:

$$F_{L,tr} = f_{AR,tr}f_{Ro,tr}C_{L,tr}\dot\psi^2\left(0.5\rho\int_0^R cr^2 \mathrm{d}r\right), \tag{15}$$

$$F_{D,tr} = f_{AR,tr}f_{Ro,tr}C_{D,tr}\dot\psi^2\left(0.5\rho\int_0^R cr^2 \mathrm{d}r\right), \tag{16}$$

$$F_{rot,1} = f_\alpha f_r C_{rot,1}\left(\rho\left|\dot\psi\right|\left|\dot\alpha\right|\right)\left(\int_0^R c^2 r\mathrm{d}r\right), \tag{17}$$

$$F_{rot,2} = 2.67\rho\dot\alpha\left|\dot\alpha\right|\left|\int_{LE}^{TE} rx|x|\mathrm{d}x, \tag{18}$$

$$F_a = f_{\lambda,\alpha}f_{AR,\alpha}f_a\frac{\rho\pi}{4}\left(\ddot\psi\sin\alpha\int_0^R c^2 r\mathrm{d}r + \ddot\alpha\int_0^R \frac{c^3}{2}\mathrm{d}r\right), \tag{19}$$

where $\rho$ is the air density, $r$ is the distance from the flapping axis to any radial position along the wingspan, $c$ is the local chord length at the given radial position $r$, $x$ is the chordwise position from the LE to the TE. Lee et al. [26] presented detailed expressions for the coefficients in the above equations as follows:

$$C_{L,tr} = (1.966 - 3.94Re^{-0.429})\sin(2\alpha), \tag{20}$$

$$C_{D,tr} = 0.031 + 10.48Re^{-0.764} + (1.873 - 3.14Re^{-0.369})(1 - \cos(2\alpha)), \tag{21}$$

$$C_{rot,1} = 0.842 - 0.507Re^{-0.1577}, \tag{22}$$

$$f_{AR,tr} = 32.9 - 32.0AR^{-0.00361}, \tag{23}$$

$$f_{Ro,tr} = -0.205\arctan(0.587(Ro - 3.105)) + 0.870, \tag{24}$$

$$f_\alpha = \begin{cases} 1 & , -45° < \alpha < 45° \\ -1 & , 135° < \alpha < 225° \\ \sqrt{2}\cos\alpha & , \text{otherwise} \end{cases}, \tag{25}$$

$$f_r = 1.570, \tag{26}$$

$$f_{\lambda,\alpha} = 47.7\lambda^{-0.0019} - 46.7, \tag{27}$$

$$f_{AR,\alpha} = 1.294 - 0.590AR^{-0.662}, \tag{28}$$

$$f_a = 0.776 + 1.911Re^{-0.687}, \tag{29}$$

where $Ro$ is defined as $Ro = R_2/c$.

As the sweeping and pitching motion are both active movements in this study, the instantaneous aerodynamic power required to drive the wing is expressed as:

$$P_{aero} = -(M_\psi\dot{\psi} + (M_\alpha + T_\alpha)\dot{\alpha}), \tag{30}$$

where $M_\psi$ and $M_\alpha$ are aerodynamic moments about flapping and pitching axis respectively, $T_\alpha$ is the torque due to added moment of inertia. As the aerodynamic power are related to the sweeping angular velocity and pitching angular velocity, the aerodynamic power can be divided into sweeping power and pitching power, which correspond to the first and second terms in Equation (30), respectively. Each aerodynamic moment term is given as:

$$M_\psi = -F_{D,tr}R_2(0.0784\cos(2\alpha) + 1.088) - 0.993R_2(F_{rot,1} + F_{rot,2})\sin\alpha - 1.078R_2F_a\sin\alpha, \tag{31}$$

$$M_\alpha = -(F_{L,tr}\cos\alpha + F_{D,tr}\sin\alpha)(-0.0799\cos(2\alpha) + 0.377)\bar{c} - 0.398\bar{c}(F_{rot,1} + F_{rot,2}) - 0.5\bar{c}F_a, \tag{32}$$

$$T_\alpha = -(1.114 + 7.89Re^{-0.855})\ddot{\alpha}\frac{\pi\rho}{128}\int_0^R c^4\mathrm{d}r. \tag{33}$$

The inertia power, the energy to overcome the inertia force during wing acceleration, can be expressed as:

$$P_{iner} = -(M_{\psi,iner}\dot{\psi} + M_{\alpha,iner}\dot{\alpha}), \tag{34}$$

where $M_{\psi,iner}$ and $M_{\alpha,iner}$ are inertia moments about the flapping and pitching axis respectively. Meanwhile, as the wing is regarded as a rigid plate, the energy stored in the elastic structure was ignored.

Different from other research on evaluating force coefficient, dimensional aerodynamic force is given more attention in this study. The major purpose of this study is to explore the degree of influence of geometric and kinematic parameters on aerodynamic performance and to identify which parameters are the dominant factors. The influence degree of these parameters may not be fully demonstrated only by the coefficient. Because the calculation process of force coefficient naturally excludes the influence of some parameters such as flapping frequency and wing area. Another purpose is to establish a more accurate and intuitive functional relationship between aerodynamic performance and the parameters than that proposed by Ellington [10]. Thus, dimensional aerodynamic force is more feature than force coefficient in this study.

The aerodynamic model mentioned above can evaluate instantaneous force and power under a specific flapping trajectory. For sensitivity analysis, the cycle-averaged force and power consumption are the objects which need to be concerned in this study, and can be calculated as:

$$L = \frac{1}{T}\int_0^T F_L(t)\mathrm{d}t, \tag{35}$$

$$P = \frac{1}{T}\int_0^T (P_{iner}(t) + P_{aero}(t))\mathrm{d}t. \tag{36}$$

According to the study of Truong et al. [25] and Phan et al. [38], the inertia force presents a minor contribution to the average resultant vertical force of hovering flexible wing, although it will increase at high frequency. Theoretically, for a rigid wing with perfectly periodic flapping trajectory, the cycle-averaged inertia power will be zero [10,39]. So, the inertia force and corresponding power are not considered in this study, which is also adopted by other researchers [40,41]. Then, the power consumption can be written as:

$$P = \frac{1}{T} \int_0^T P_{aero}(t)\mathrm{d}t, \tag{37}$$

Consequently, only the time-averaged aerodynamic power will be discussed in the subsequent sections.

Power loading (lift per unit power) is defined as:

$$PL = \frac{L}{P}. \tag{38}$$

## 3. Results and Discussion

### 3.1. Effect of Wing Geometric Parameters

The flapping kinematics of the wing are kept constant when performing the sensitivity analysis for the geometric parameters. The motion of the wing consists of sweeping and pitching movements, and the main kinematic parameters are shown in Table 4.

**Table 4.** Kinematic parameters used in sensitivity analysis for the geometric parameters.

| Parameter | Description | Value |
|---|---|---|
| $\psi_m$ | Sweeping amplitude (degree) | 60 |
| $\theta_m$ | Pitching amplitude (degree) | 45 |
| $\varphi$ | Phase shift of pitching motion (degree) | 90 |
| $\theta_0$ | Pitching motion offset (degree) | 90 |
| $K$ | Affects of the shape of $\psi$ | 0.01 |
| $C_\theta$ | Affects of the shape of $\theta$ | 4.00 |

### 3.1.1. Aspect Ratio $AR$

The aerodynamic performance of wings with different $AR$ was evaluated and compared under various flapping frequencies. The frequency simulated ranges from 10 Hz to 30 Hz and only the results of four specific frequencies are presented. In fact, the results of all frequencies show the same trend. As can be seen in Figure 4, the cycle-averaged lift force and aerodynamic power increase monotonically as $AR$ increases. Additionally, the increase of lift appears to be gentler when the $AR$ is large, e.g., $AR > 5$. According to the study by Shahzad et al. [11], the reason for this might be the detachment of the vortex structure on the upper surface at higher $AR$, leading to a loss of lift. The growth rate of the curves which means the slope is larger at higher frequency. It seems that the $AR$ presents a linear relationship with lift and power. However, the increase in $AR$ shows a disadvantage in aerodynamic efficiency. Power loading decreases gradually with the increase of $AR$. The reason for this might be that, at higher $AR$, flapping motion consumes more significant power, even if the lift generated is considerable. This phenomenon can also explain why a lower power loading appears at a higher flapping frequency.

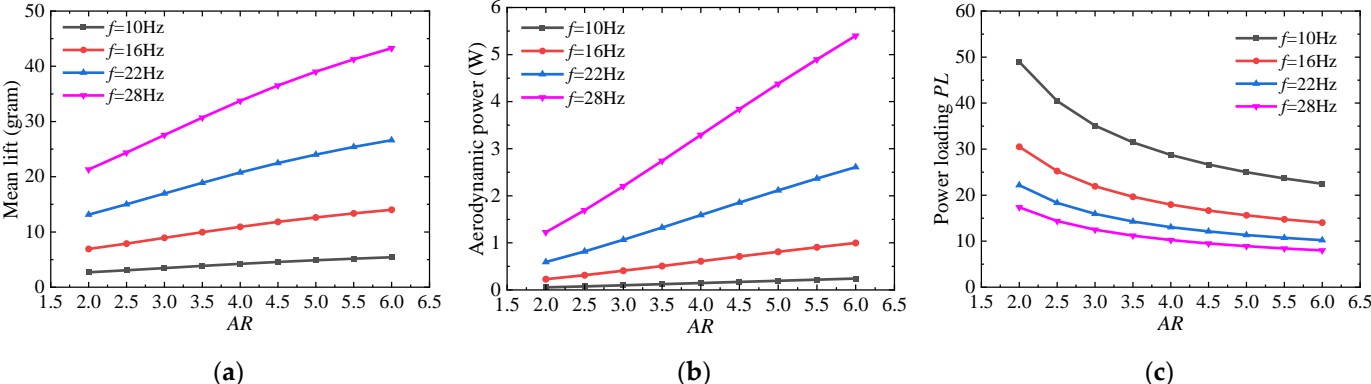

**Figure 4.** Aerodynamic performance of wings with different *AR*. (**a**) Mean lift of different *AR*. (**b**) Aerodynamic power of different *AR*. (**c**) Power loading of different *AR*.

To further investigate the effect of *AR* on the aerodynamic performance of the flapping wing, sensitivity analysis was performed at several frequencies. Wing 3 in Group 1 where the *AR* equals 3 was selected as the base state. The relationship between *AR* variation and lift variation is established and presented in Figure 5, as well as the power and *PL*. Interestingly, the flapping frequency shows a minor impact on the sensitivity analysis result, indicating the kinematic parameters are independent of the geometric parameters for the rigid wing. However, in reality, due to the flexibility of the wing, the kinematic parameters are more or less potentially related to the geometric parameters. For example, with the increase of flapping frequency, the flexible deformation of wing surface turns to be obvious gradually. Subsequently, the aerodynamic performance will be affected. Under these circumstances, the results of the sensitivity analysis for a certain geometric parameter might be related to kinematic parameters. Therefore, the results in this study might be limited in the application of flapping wings with high rigidity.

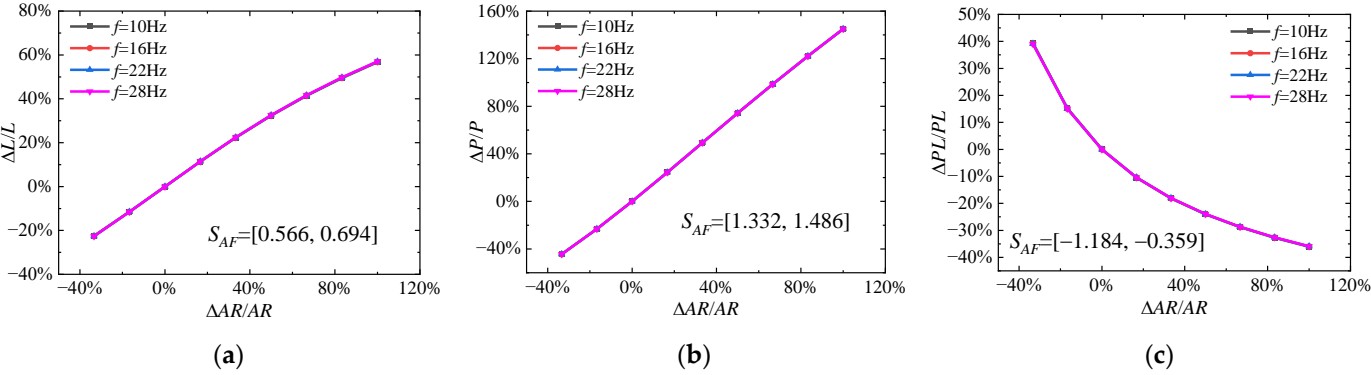

**Figure 5.** Sensitivity analysis for *AR*. The value of $S_{AF}$ is given as an interval between the minimum and maximum values. (**a**) Variation of lift vs. variation of *AR*. (**b**) Variation of aerodynamic power vs. variation of *AR*. (**c**) Variation of *PL* vs. variation of *AR*.

As presented in Figure 5, an approximately linear relationship can be clearly found between *AR* and mean lift as well as aerodynamic power. The *PL* is negatively correlated with *AR*. Meanwhile, the increase of *AR* does not seem to cause a significant decrease in power loading anymore.

Based on Equations (1)–(3), the aerodynamic performance as a function of sensitivity parameter could be described as a power function:

$$y = bx^a, \tag{39}$$

where *b* is the coefficient, and *a* is the exponent. Thus, *a* and *b* can be easily identified by linear fitting on log-transformed data.

Figure 6 demonstrates the encouraging results that the data after log-transformed presents a dramatic linear correlation. Table 5 presents the coefficients for the power relationship between aerodynamic performance and *AR* at various flapping frequencies. It can be found that, same as the expression in Equations (1)–(3), the flapping frequency will affect the constant *b* but shows little impact on the exponent *a* which is the more important variable. Meanwhile, the exponent value obtained by linear fitting is closed to the theoretical value, although there are some deviations. According to the simplification by Ellington, Equations (1)–(3) mainly account for the dominant force formed by the translation mechanism without considering the rotation and added mass mechanisms, which may be the reason for the deviations.

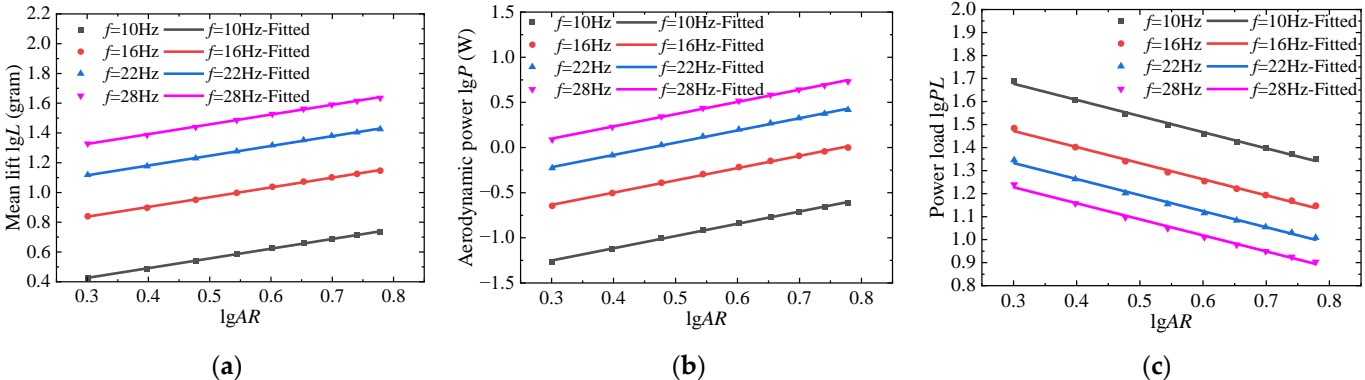

(a)                                    (b)                                    (c)

**Figure 6.** Logarithmic relationship of aerodynamic performance with *AR*. Symbols are raw data, and solid lines are fitted data. (**a**) Mean lift with *AR*. (**b**) Aerodynamic power with *AR*. (**c**) Power loading with *AR*.

**Table 5.** Coefficients for the power relationship between aerodynamic performance and *AR* at different flapping frequencies.

| Frequency | *L* | | | *P* | | | *PL* | | |
|---|---|---|---|---|---|---|---|---|---|
| | *a* | *b* | *R*-Squared | *a* | *b* | *R*-Squared | *a* | *b* | *R*-Squared |
| 10 Hz | 0.654 | 1.697 | 0.999 | 1.356 | 0.022 | 0.998 | −0.702 | 77.256 | 0.995 |
| 16 Hz | 0.657 | 4.367 | 0.999 | 1.356 | 0.091 | 0.998 | −0.699 | 48.084 | 0.995 |
| 22 Hz | 0.658 | 8.281 | 0.999 | 1.356 | 0.237 | 0.998 | −0.698 | 34.877 | 0.995 |
| 28 Hz | 0.659 | 13.440 | 0.999 | 1.356 | 0.491 | 0.998 | −0.697 | 27.351 | 0.995 |

Consistent with the previous discussion, for the rigid flapping wing, the flapping frequency has little effect on the sensitivity of geometric parameters. By referring to the exponent value presented in Table 5, the approximate relationship between aerodynamic performance and *AR* can be obtained and written as follows:

$$L \propto AR^{0.659}; \ P \propto AR^{1.356}; \ PL \propto AR^{-0.697}. \tag{40}$$

### 3.1.2. Wing Area *S*

Figure 7 presents the aerodynamic performance of wings with different surface area *S*. Different from the result of *AR*, the wing surface area *S* presents a more allometric effect on lift and power. With the increase of *S*, the lift will increase exponentially, as well as the power consumption. Whereas the power loading decreases quickly at first and then more gradually, suggesting the growth ratio of power is much higher than that of lift. At larger wing area *S*, the increase in lift seems comparable with the variation of power, resulting in a slow decrease in power loading. This phenomenon can also be seen in the result of *AR* in Figure 5c.

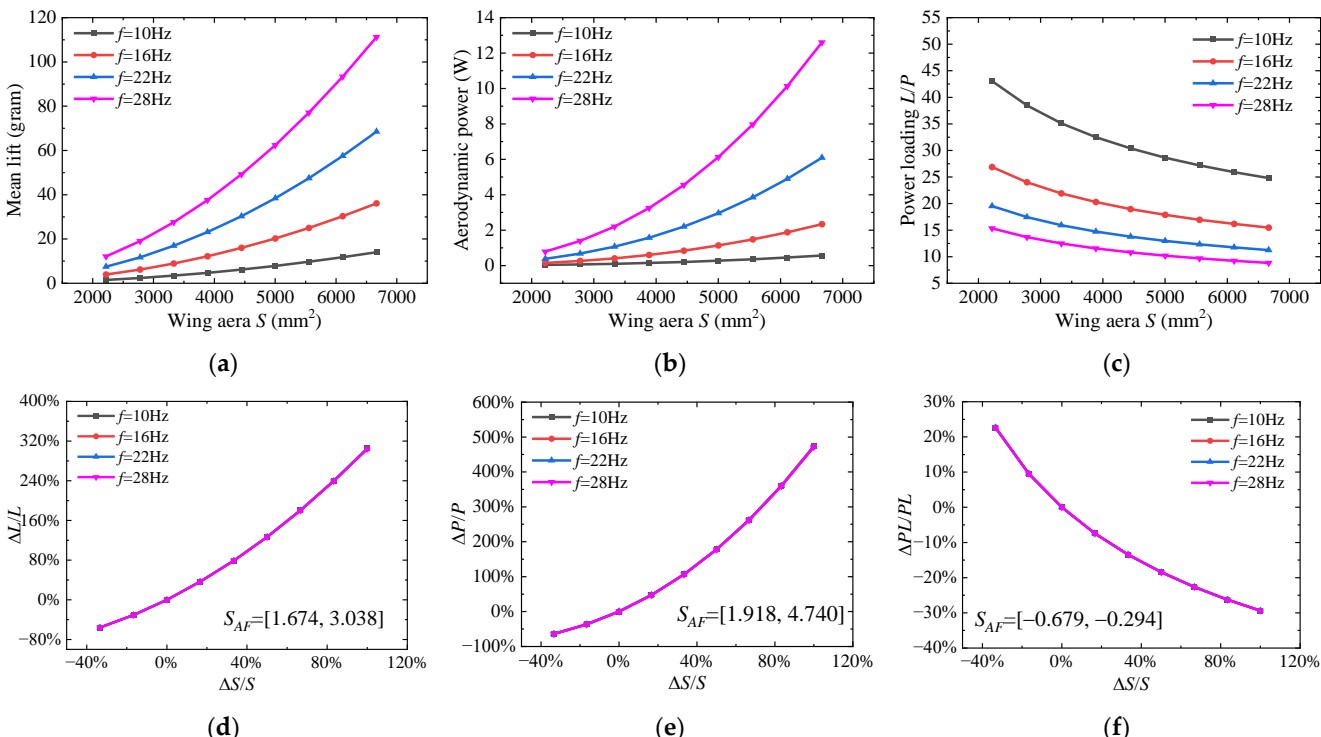

**Figure 7.** Sensitivity analysis for wing surface area *S*. (**a**) Mean lift of different *S*. (**b**) Aerodynamic power of different *S*. (**c**) Power loading of different *S*. (**d**) Variation of lift vs. variation of *S*. (**e**) Variation of aerodynamic power vs. variation of *S*. (**f**) Variation of *PL* vs. variation of *S*.

To conduct the sensitivity analysis for wing area *S*, wing 12 in Group 2 is selected as the basis wing. The analysis results are presented in Figure 7d–f. Apparently, flapping frequency is also independent of the *S* variation, since the results of different frequencies are almost consistent. The influence of *S* on lift and power is nonlinear. The larger the increment of wing area, the higher the increment of lift and power consumption, but the slower the decrease of power loading.

Linear fitting was also performed on the log-transformed data to obtain the relationship between *S* and corresponding aerodynamic performance. A similar linear relationship can also be found in Figure 8, indicating that wing surface area *S* also satisfies the exponential assumption in Equation (39). Table 6 gives the exponent values obtained by linear fitting. It can be found that the exponent values are very close to the theoretical result in Equations (1)–(3). The constant *b*, the intercept of the fitted line, is related to the flapping frequency, which is not presented here.

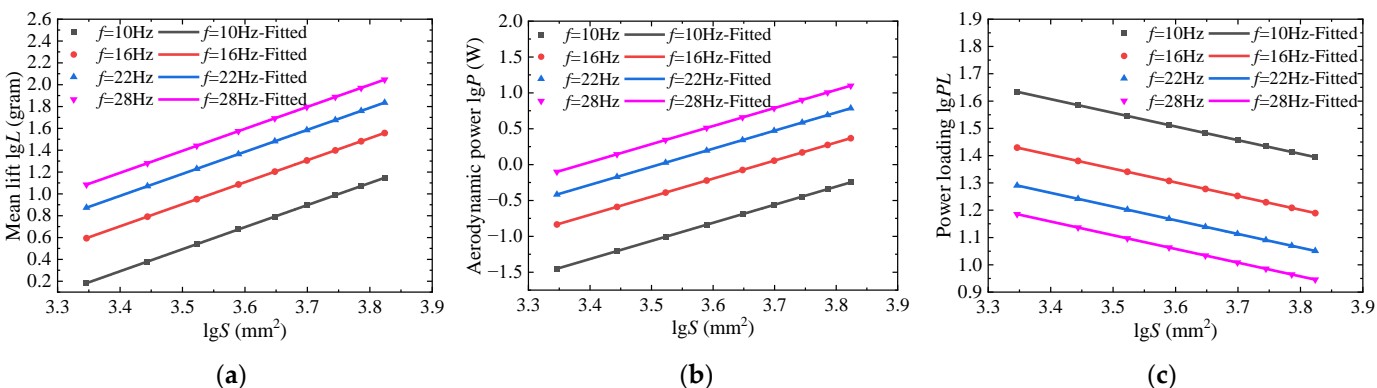

**Figure 8.** Logarithmic relationship of aerodynamic performance with *S*. Symbols are raw data, and solid lines are fitted data. (**a**) Mean lift with *S*. (**b**) Aerodynamic power with *S*. (**c**) Power loading with *S*.

**Table 6.** Coefficients for the power relationship between aerodynamic performance and $S$ at different flapping frequencies.

| Frequency | L | | P | | PL | |
|---|---|---|---|---|---|---|
| | $a$ | R-Squared | $a$ | R-Squared | $a$ | R-Squared |
| 10 Hz | 2.021 | 0.999 | 2.523 | 0.999 | −0.502 | 0.999 |
| 16 Hz | 2.018 | 0.999 | 2.520 | 0.999 | −0.502 | 0.999 |
| 22 Hz | 2.016 | 0.999 | 2.518 | 0.999 | −0.502 | 0.999 |
| 28 Hz | 2.015 | 0.999 | 2.517 | 0.999 | −0.502 | 0.999 |

According to Table 6, the approximate relationship between aerodynamic performance and S can be obtained and written as follows:

$$L \propto S^{2.015}; \ P \propto S^{2.517}; \ PL \propto S^{-0.502}. \tag{41}$$

3.1.3. The Dimensionless Radius of the Second Moment of Area $r_2$

The dimensionless radius of the second moment of area $r_2$ has been demonstrated to be an important factor in hovering flight aerodynamic performance. Research by Weis-Fogh [42] has shown that the mean lift force is proportional to the second moment of wing area in the quasi-steady state. According to the statistical data of insect wings' morphological parameters by Ellington [43], $r_2$ of natural insect wings mostly ranges from 0.5 to 0.6. a larger $r_2$ means the wing area concentrates towards the wing tip, such as the forewing of *Chrysoperla carnea*. Conversely, lower $r_2$ indicates the centroid of the area is closer to the wing root, such as the hindwing of *Manduca sexta*.

In light of the important role of $r_2$ in the aerodynamic performance of hovering wing, it was systematically investigated with four groups of wings as present in Tables 2 and 3. Three ways of modulating $r_2$ are introduced, namely changing the magnitude of the maximum chord length, the location of the maximum chord length, and the taper ratio. It should be noted that the taper ratio was kept constant when the maximum chord length was the research object.

First, for the wings in Group 3 with different $r_2$ by changing the location of maximum chord length $c_{max}$, $c_{max}$ remains constant and equals 1.4 times root chord length. Wing 22 in Group 3 is selected to calculate the sensitivity. According to Figure 9, a strong linear relationship can be found between $r_2$ and mean lift at any frequency, as well as aerodynamic power. However, $r_2$ is only able to vary −3% to 5% by modulating the location of $c_{max}$, and the variation of lift and power seems slight. Nevertheless, the sensitivity of $r_2$ under this circumstance is relativity higher than that of $AR$ by referring to Figure 5, suggesting a more significant impact of $r_2$ on the aerodynamic performance. The result of power loading is not presented here and will be discussed later.

Then, focus on the $r_2$ by modulating the magnitude of $c_{max}$. It is worth noting that increasing the chord length at the wing root will lead to the concentration of area toward the wing root and a decrease of $r_2$. However, the increase of chord length near the wing tip will cause a distal concentration of the wing and an increase of $r_2$. It can be found that the modulation of $c_{max}$ at different locations will lead to different results of $r_2$. Therefore, two spanwise locations, 0.25$R$ and 0.75$R$, are selected when the magnitude of $c_{max}$ is adjusted, corresponding to the wings in Group 4 and Group 5 respectively.

Wing 30 and wing 39 are chosen as the base state in each group to calculate the sensitivity and the results are presented in Figure 10. In all cases, the positive $S_{AF}$ value indicates that with the increase of $r_2$, the lift and power gradually increase. Regardless of the spanwise location, the linear correlation between lift and $r_2$ is less obvious than that between power and $r_2$. Additionally, a higher $S_{AF}$ value shows that $r_2$ has a much greater impact on power than lift. Considering the results at different spanwise locations, it can be found that the variation of $r_2$ at the wing tip presents a greater influence on the force and power, because a higher $S_{AF}$ is obtained at 0.75$R$. This phenomenon suggests that the

increase of chord length near the wing tip might offer an advantage on lift generation, while it consumes more power.

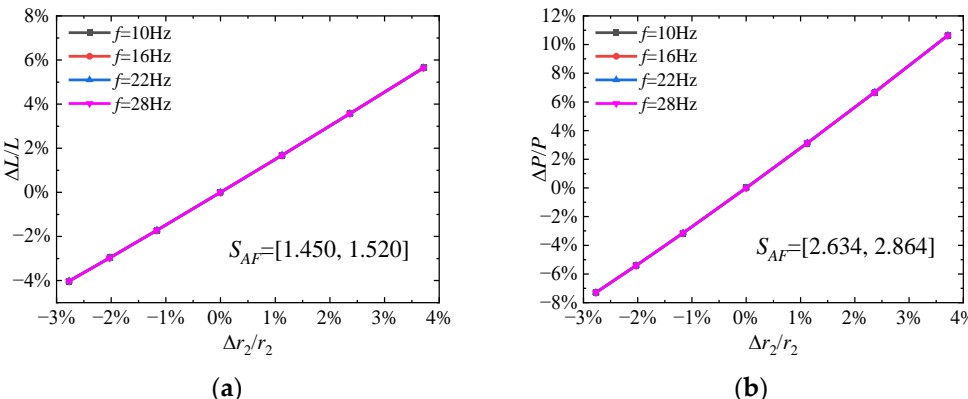

(a)                                            (b)

**Figure 9.** Sensitivity analysis for the dimensionless radius of the second moment of area $r_2$ by changing the location of $c_{max}$. (**a**) Variation of lift vs. variation of $r_2$. (**b**) Variation of aerodynamic power vs. variation of $r_2$.

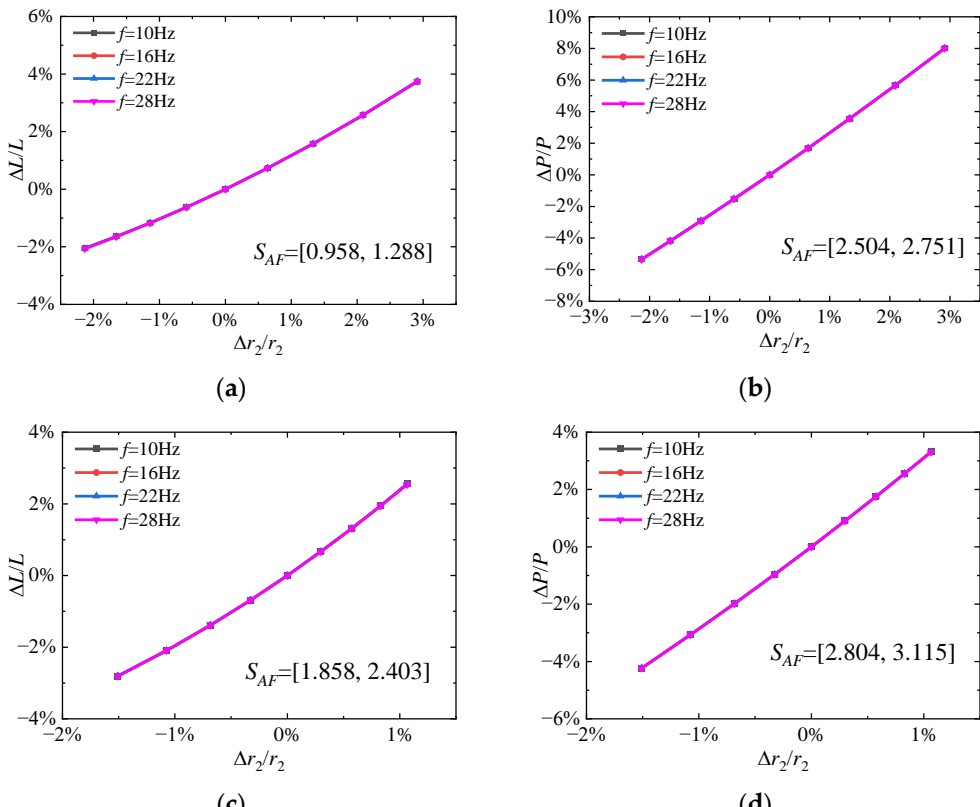

(a)                                            (b)

(c)                                            (d)

**Figure 10.** Sensitivity analysis for the dimensionless radius of the second moment of area $r_2$ by modulating the value of $c_{max}$ at different spanwise locations. (**a**) Variation of lift vs. variation of $r_2$, modulating $c_{max}$ at $0.25R$. (**b**) Variation of aerodynamic power vs. variation of $r_2$, modulating $c_{max}$ at $0.25R$. (**c**) Variation of lift vs. variation of $r_2$, modulating $c_{max}$ at $0.75R$. (**d**) Variation of aerodynamic power vs. variation of $r_2$, modulating $c_{max}$ at $0.75R$.

One interesting question is, since $r_2$ varies with the modulation of both the magnitude and location of $c_{max}$, which is the better strategy to modulate $r_2$? Figure 11a depicts the relationship between $r_2$ and the magnitude of $c_{max}$ as well as the location. As mentioned above, a higher $r_2$ can be obtained at a larger $c_{max}$ located near the wing tip. Considering

the proportional relationship between lift and $r_2$, a higher lift appears at a higher $r_2$ as shown in Figure 11b. Consequently, the way which can achieve a larger $r_2$ variation, i.e., a higher lift increment, is the primary factor to be considered.

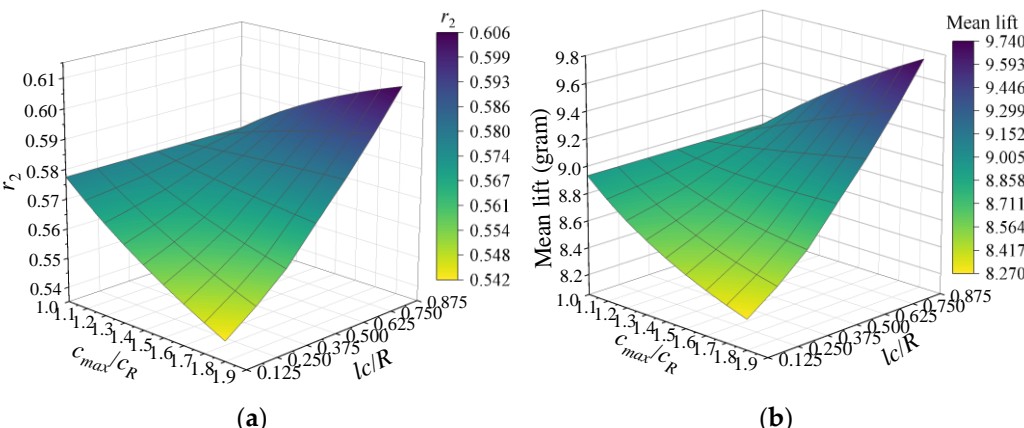

**(a)**         **(b)**

**Figure 11.** (**a**) The dimensionless radius of the second moment of area $r_2$ and (**b**) lift performance with respect to the magnitude and location of the maximum chord length $c_{max}$ at the flapping frequency of 16 Hz.

A similar sensitivity analysis was performed for $r_2$ by tuning the magnitude and location of $c_{max}$ with small variation. As shown in Figure 12, a higher and positive variation of $r_2$ can also be obtained with a larger $\Delta c_{max}$ near the wing tip, which is similar to the result in Figure 11a. A positive and larger boundary curve slope indicates that an obvious increment of $r_2$ can be obtained. Apparently, a rapid growth of $r_2$ occurs as the location of the maximum chord length moves towards the wing tip. Whereas the increase of $c_{max}$ can lead to the increase of $r_2$, the growth rate of $r_2$ is not as fast as adjusting the location of $c_{max}$. Therefore, if one wants to improve hovering lift by adjusting $r_2$, then shifting the location of the maximum chord length towards the wing tip might be a better choice.

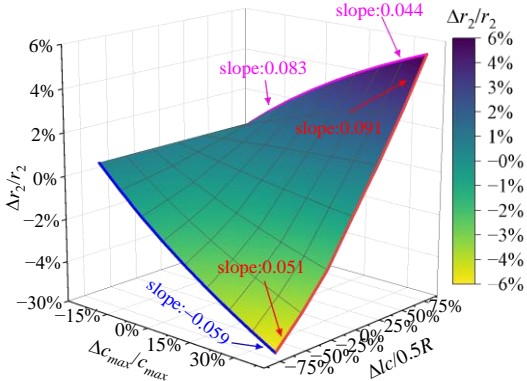

**Figure 12.** Sensitivity analysis for $r_2$. Base state: $c_{max}/c_R = 1.4$, location of the maximum chord length $lc = 0.5R$, $r_2 = 0.574$.

Finally, various $r_2$ with different taper ratios $\lambda$ are considered, corresponding to the wings in Group 6. In this part, $r_2$ was tuned by varying taper ratio $\lambda$. A larger $\lambda$ means a larger tip chord length and a shorter root chord length, resulting in a concentration of area toward the wing tip and a larger $r_2$. The base wing for the sensitivity analysis in this part is wing 48 which is a rectangular planform.

As can be seen from Figure 13a, a large variation of $\lambda$ can only achieve a slight increase in lift of about 10%. The smaller $S_{AF}$ value indicates that the influence of $\lambda$ on lift is less obvious than that of other parameters, although the lift increases with an increase in $\lambda$. To

some extent, changing $\lambda$ is the synchronous adjustment of magnitude and location of $c_{max}$. Therefore, it may be reasonable that changing $\lambda$ can achieve a more pronounced variation in $r_2$ compared to other methods.

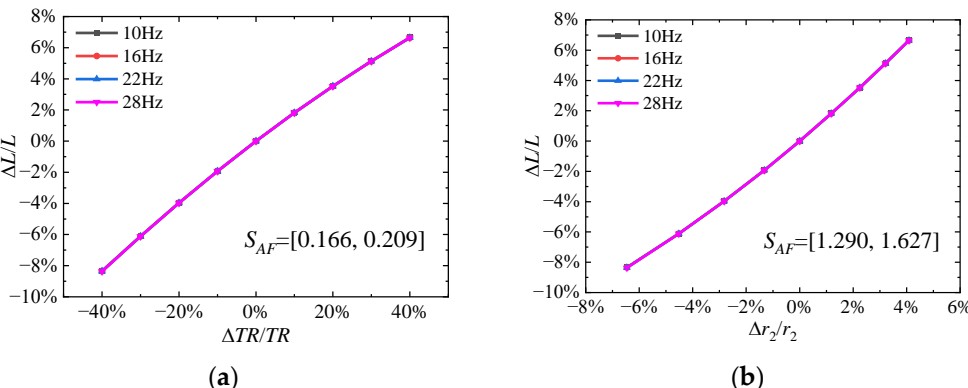

**Figure 13.** Sensitivity analysis for the dimensionless radius of the second moment of area $r_2$ by modulating the taper ratio $\lambda$. (**a**) Variation of lift vs. variation of $\lambda$. (**b**) Variation of lift vs. variation of $r_2$.

Since the aforementioned $r_2$ can be generated through different methods, it is worth discussing whether all these $r_2$ can still present a consistent relationship with the aerodynamic performance, no matter which way is chosen to modulate $r_2$. Therefore, all the $r_2$ were reordered with the corresponding force and power. Meanwhile, the sensitivity analysis was also calculated based on the rearranged data.

As shown in Figure 14, it is encouraging to note that, a consistent phenomenon still exists in most cases between $r_2$ and the lift and power, as well as the power loading, i.e., the larger the $r_2$, the greater the lift and power, and the smaller the power loading. Nevertheless, there are still some slight fluctuations that can be found from the reordered data, especially the lift curves in Figure 14a,d. To conduct the sensitivity analysis based on the oscillating data, an optimal quadratic polynomial function is used to fit the corresponding data. Therefore, the trend of aerodynamic performance varying with $r_2$ can also be reflected by the fitted curve. Apparently, within the range of $r_2$ study in this section, a nonlinear relationship can be found between $r_2$ and lift, which is consistent with the estimated relationship in Equation (1). With the increase of $r_2$, the concentration of wing area moves towards the wing tip. The larger area with a higher sweep velocity will generate more drag force near the tip region along the flapping direction, which will lead to a rapid increase in aerodynamic power. A gradual decrease of power loading also indicates that $r_2$ has a more obvious effect on aerodynamic power than lift.

Figure 15 shows the logarithmic relationship established based on the reordered $r_2$ and corresponding aerodynamic performance. Although the explicit expression for the power and power loading with $r_2$ is not given in Equations (2) and (3), a significant linear relationship can still be observed in Figure 15 and Table 7 on the log-transformed data. This phenomenon indicates that an exponential relationship can be established between the power and power loading with $r_2$. Similar to the aforementioned study, the exponent $a$ basically does not vary with the flapping frequency, whereas the constant $b$ will change with the frequency.

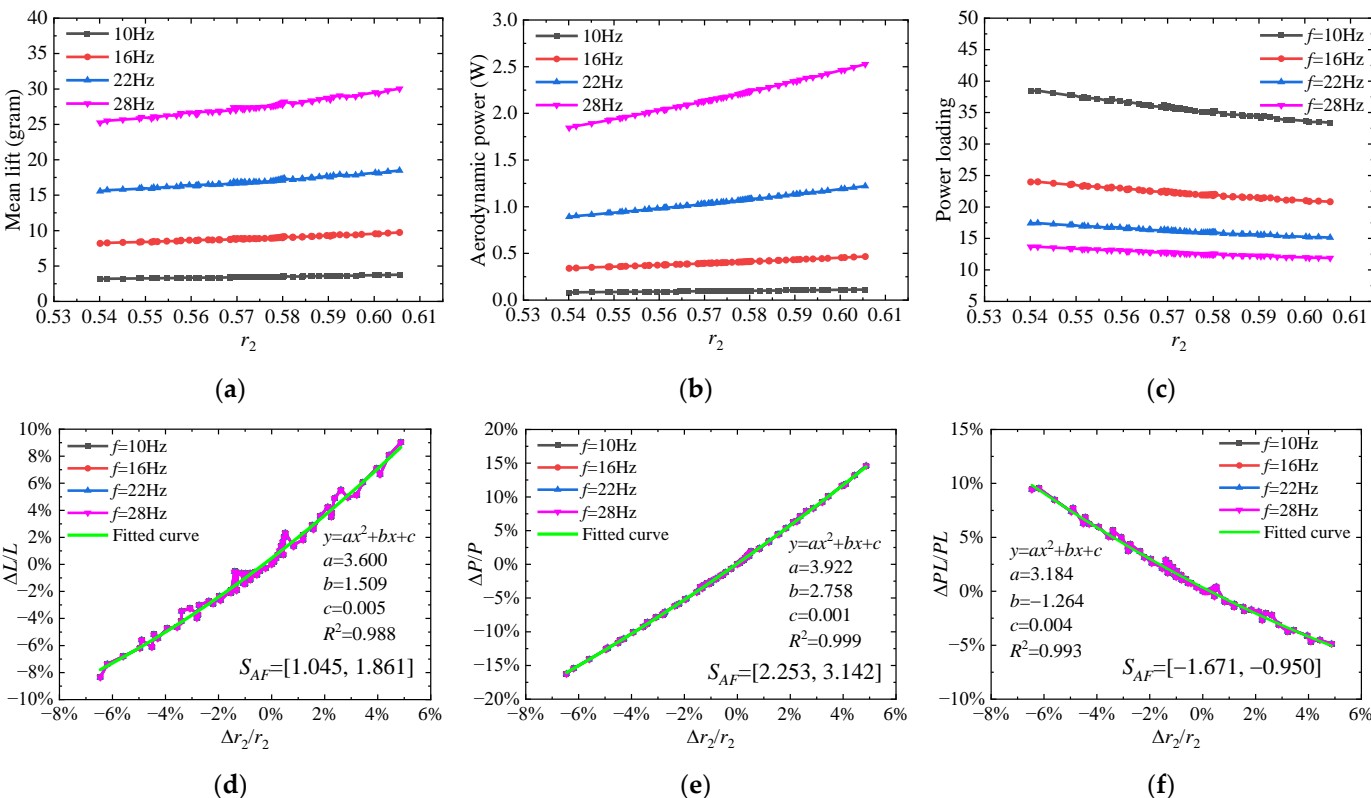

**Figure 14.** Sensitivity analysis for the dimensionless radius of the second moment of area $r_2$ base on the reordered data. All the $r_2$ were reordered with the corresponding force and power. (**a**) Mean lift of different $r_2$. (**b**) Aerodynamic power of different $r_2$. (**c**) Power loading of different $r_2$. (**d**) Variation of lift vs. variation of $r_2$. (**e**) Variation of aerodynamic power vs. variation of $r_2$. (**f**) Variation of power loading vs. variation of $r_2$.

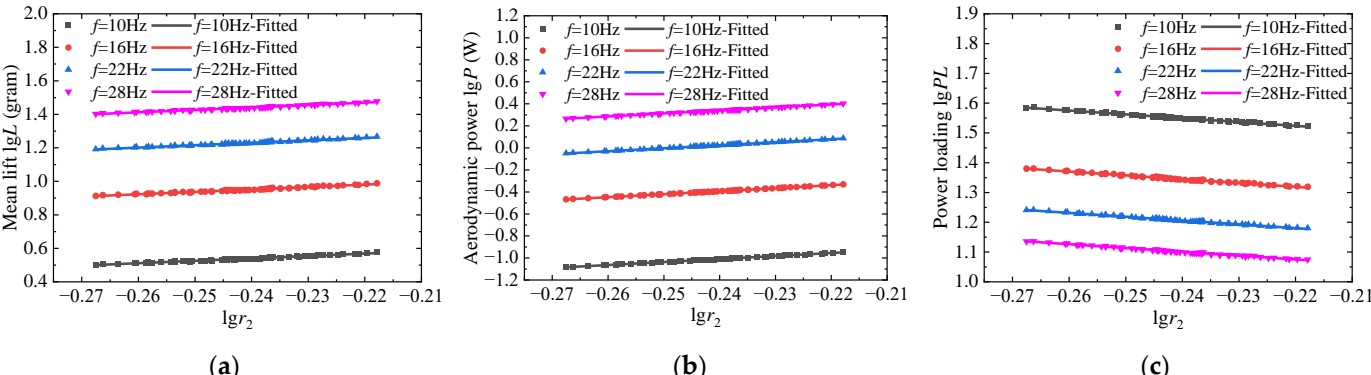

**Figure 15.** Logarithmic relationship of aerodynamic performance with $r_2$. Symbols are raw data, and solid lines are fitted data. (**a**) Mean lift with $r_2$. (**b**) Aerodynamic power with $r_2$. (**c**) Power loading with $r_2$.

**Table 7.** Coefficients for the power relationship between aerodynamic performance and $r_2$ at different flapping frequencies.

| Frequency | $L$ | | $P$ | | $PL$ | |
|---|---|---|---|---|---|---|
| | $a$ | $R$-Squared | $a$ | $R$-Squared | $a$ | $R$-Squared |
| 10 Hz | 1.441 | 0.982 | 2.724 | 0.999 | −1.279 | 0.991 |
| 16 Hz | 1.442 | 0.983 | 2.723 | 0.999 | −1.280 | 0.991 |
| 22 Hz | 1.442 | 0.983 | 2.723 | 0.999 | −1.281 | 0.990 |
| 28 Hz | 1.443 | 0.983 | 2.722 | 0.999 | −1.283 | 0.990 |

According to Table 7, the approximate relationship between aerodynamic performance and $r_2$ can be obtained and written as follows:

$$L \propto r_2^{1.443}; \; P \propto r_2^{2.722}; \; PL \propto r_2^{-1.283}. \tag{42}$$

### 3.1.4. Summary of the Sensitivity Analysis for Wing Geometric Parameters

A sensitivity analysis shows that the aspect ratio *AR*, wing area *S*, and second moment of area $r_2$ all have a significant impact on the hovering aerodynamic performance. It is necessary to evaluate and compare the degree of influence of each geometric parameter, which in turn helps the designer select the dominant factor to enhance the performance. Therefore, the sensitivity analysis results for the three parameters are compared in Figure 16. One of the main differences is that due to the limitation of the designed samples in this study, it is hard to fully keep an identical variation range of independent variables. For example, the variation of $r_2$ is much smaller than that of the other two parameters. However, the influence degree of each parameter can also be identified by the slope of the sensitivity curves.

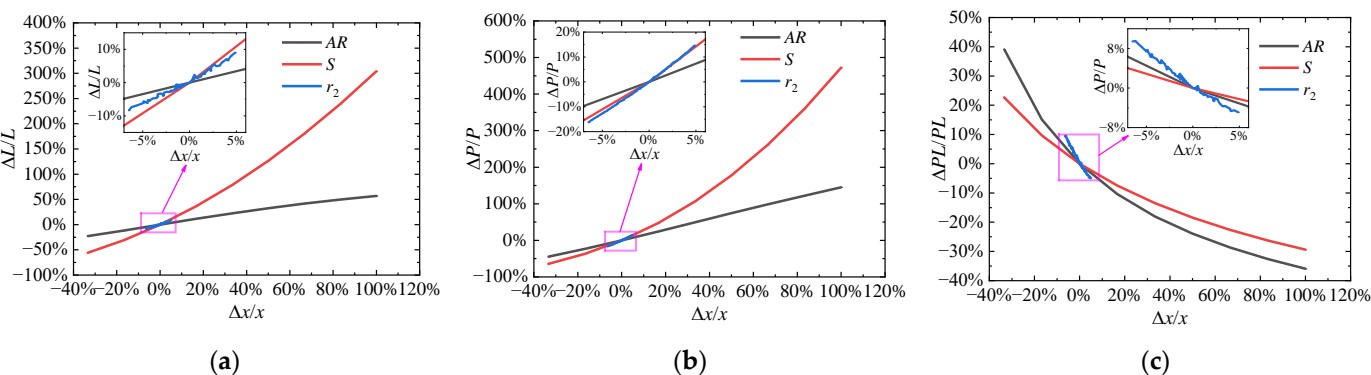

(a)    (b)    (c)

**Figure 16.** Comparison of sensitivity analysis results for the geometric parameters. $\Delta x$ is the variation of the corresponding independent variable, e.g., $\Delta S$ for *S*. (**a**) Sensitivity of lift. (**b**) Sensitivity of aerodynamic power. (**c**) Sensitivity of power loading.

Concerning the impact on lift, it can be clearly seen from Figure 16a that wing area *S* has the most significant effect on the lift increment, followed by $r_2$ and finally *AR*. Consistent with the exponent value in Table 8, lift is scaled with $S^{2.015}$ and $r_2^{1.443}$. The impact of $r_2$ and *S* on aerodynamic power is essentially the same for very close exponent values of *S* and $r_2$. Figure 16b shows that $r_2$ seems to be the dominant factor in power consumption when the three parameters vary within the same variation of $r_2$, corresponding to the highest exponent value in Table 8. However, when the increment of the variable is larger, wing area *S* might lead to a more significant increase in power, as the area curve grows more rapidly than others. From Figure 16c and Table 8, it can be clearly seen that the increase of all three parameters is detrimental to the power loading. More importantly, increasing $r_2$ brings the most rapid decrease in power loading. In contrast, drop in power loading caused by the increase of wing area appears to be gentler than other variables. As a result, increasing the wing area *S* may be a more economical option to enhance lift because the decline in power loading is not as quick.

**Table 8.** Comparison of the exponent *a* of different sensitivity parameters at 28 Hz.

|       | *L*   | *P*   | *PL*    |
|-------|-------|-------|---------|
| *AR*  | 0.659 | 1.356 | −0.697  |
| *S*   | 2.015 | 2.517 | −0.502  |
| $r_2$ | 1.443 | 2.722 | −1.283  |

During the sensitivity analysis process, the sample of the parameters follows the principle of controlling variables. Only one parameter is changed at a time, while the remaining parameters are maintained constant. In other words, *AR*, *S*, and $r_2$ are designed as independent variables when conducting the analysis process. Finally, the exponential relationship shown in Table 8 was obtained. Therefore, by referring to Equations (1)–(3), the following approximate relationship can be established:

$$L \propto AR^{0.659} S^{2.015} r_2^{1.443}, \tag{43}$$

$$P \propto AR^{1.356} S^{2.517} r_2^{2.722}, \tag{44}$$

$$PL \propto AR^{-0.697} S^{-0.502} r_2^{-1.283}. \tag{45}$$

### 3.2. Effect of Wing Kinematic Parameters

In this section, wing 3 in Group 1 was selected to perform the sensitivity analysis for kinematic parameters. Unless mentioned otherwise, the tests were carried out at a moderate flapping frequency of 20 Hz.

#### 3.2.1. Flapping Frequency and Sweeping Amplitude

As discussed in previous studies [20,44,45], the flapping frequency and sweeping amplitude, as key kinematic parameters, play a vital role in the hovering performance of flapping wing. Given that both parameters have an impact on aerodynamic performance, it is worthwhile to compare the degree of influence of these two parameters. The results of the sensitivity analysis for the flapping frequency and sweeping amplitude are given in Figure 17. Other kinematic parameters except the sweeping amplitude and flapping frequency remain constant values as presented in Table 4. The sweeping amplitude ranges from 30° to 90°, and the baseline for the sensitivity analysis is 60°. The flapping frequency varies from 10 Hz to 30 Hz, and the baseline is 20 Hz. The sensitivity analysis for each parameter is performed independently, which means only one parameter is changed at a time and the other is held constant.

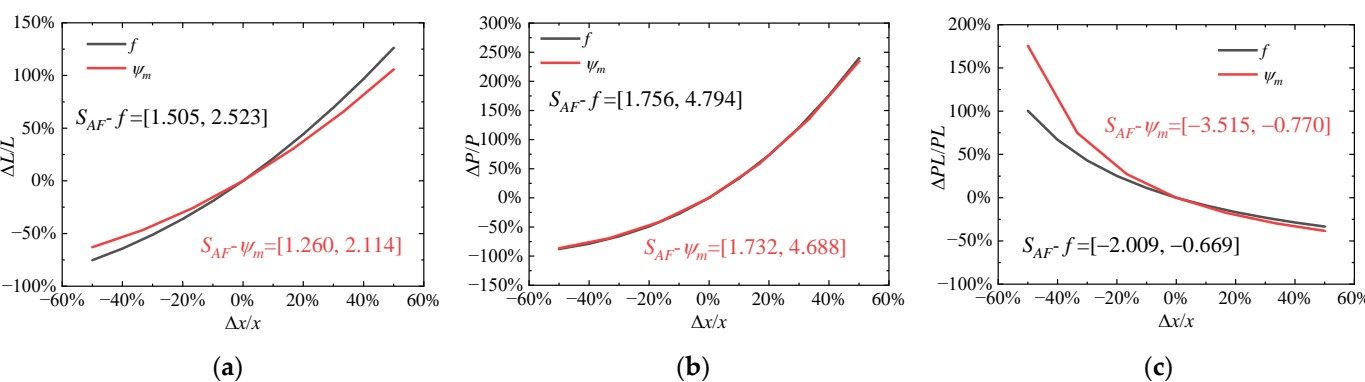

**Figure 17.** Comparison of sensitivity analysis results for the flapping frequency and sweeping amplitude. (**a**) Sensitivity of lift. (**b**) Sensitivity of aerodynamic power. (**c**) Sensitivity of power loading.

Obviously, lift and power grow monotonically with the increase of flapping frequency and sweeping amplitude, which is consistent with the expression in Equations (1) and (2). It can be seen from this study that flapping frequency results in a higher $S_{AF}$ value than sweeping amplitude, indicating that lift increases faster as frequency increases. The same phenomenon is also corroborated by the exponents in Table 9 obtained through curve fitting on the log-transformed data. However, the exponent of sweeping amplitude shows a little discrepancy with the theoretical values. Although flapping frequency and sweeping amplitude present different effects on lift, their effects on aerodynamic power are nearly identical. Moreover, the influence of frequency and sweeping amplitude on aerodynamic power is significantly greater than that on lift. In fact, according to the exponent in

Table 9, the aerodynamic power is almost related to the third power of frequency and sweeping amplitude.

**Table 9.** Coefficients for the power relationship calculated from the log-transformed data.

| Parameter | *L* | | *P* | | *PL* | |
|---|---|---|---|---|---|---|
| | *a* | *R*-Squared | *a* | *R*-Squared | *a* | *R*-Squared |
| $f$ | 2.014 | 0.999 | 3.017 | 0.999 | −1.003 | 0.999 |
| $\psi_m$ | 1.415 | 0.983 | 2.876 | 0.999 | −1.462 | 0.992 |

For power loading, a more rapid drop can be seen with an increase in sweeping amplitude, corresponding to a larger negative exponent value of the sweeping amplitude in Table 9. This phenomenon implies that increasing the sweeping amplitude to increase lift might result in a greater decrease in aerodynamic efficiency.

In the design process of the flapping wing system, modulating the frequency is much easier than changing the amplitude, which has been fixed when the design of the flapping mechanism is finished. In actual flight, adjusting the frequency is the most straightforward and achievable method to modulate the hovering lift. The sweeping amplitude is always set to a quite larger value (e.g., 90°) to maximize the lift. Another purpose is to generate additional lift through the clap and fling effect formed by the interaction of left and right wing.

To date, exponential relationships between key parameters and aerodynamic performance have been established by referring to Equations (1)–(3). Since the variables are independent of each other as previously described, the following approximate relationships can be obtained:

$$L \propto f^{2.014} \psi_m^{1.415} AR^{0.659} S^{2.015} r_2^{1.443}, \tag{46}$$

$$P \propto f^{3.017} \psi_m^{2.876} AR^{1.356} S^{2.517} r_2^{2.722}, \tag{47}$$

$$PL \propto f^{-1.003} \psi_m^{-1.462} AR^{-0.697} S^{-0.502} r_2^{-1.283}. \tag{48}$$

### 3.2.2. Pitching Amplitude and Phase Shift

Pitching motion has been observed to be a major flapping component of natural flyers, particularly for the small scale birds and insects [32,46]. Unsteady mechanisms of pitching motion, whether active or passive pitching, play important roles in force generation and flight control of flapping wings [39,47–49]. According to Dickinson et al. [50] who analyzed the aerodynamic basis of wing rotation using a dynamically scaled model of the fruit fly, pitching motion can first induce a favorable angle of attack to maximize the translational lift force. Another contribution is the rotation mechanism which can generate a noticeable force component during stroke reversal.

Pitching motion combines two key parameters, namely pitching amplitude and phase angle between sweeping and pitching motion. The pitching amplitude determines the angle of attack during wing's sweeping. The phase angle defines the timing of rotational motion during stroke reversal. Lift and power performance of two typical states are presented in Figure 18. Consistent with the previous research [23,51], aerodynamic performance, especially lift, does not vary monotonically with the pitching amplitude or phase angle. This phenomenon implies that there are some optimal values to maximize the performance. For example, when the pitching amplitude is fixed at 45°, the peak lift appears at phase angle $\varphi = 60°$, corresponding to an advance in rotation relative to sweeping. Moreover, the maximum lift can be obtained at pitching amplitude $\theta_m = 55°$ when phase angle is kept at 90°. Given the inconsistent phenomenon with the previous kinematic parameters, the cooperation effect of pitching amplitude and phase angle on the aerodynamic performance is investigated in this section.

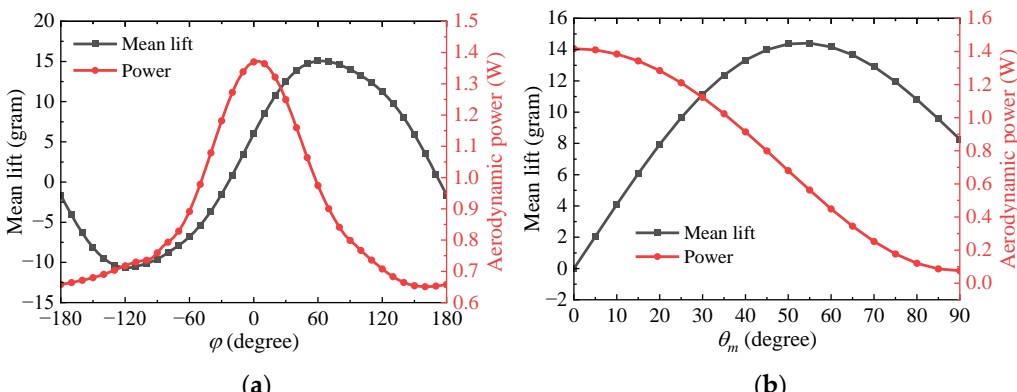

**Figure 18.** Aerodynamic performance of two typical states. (**a**) Lift and power with different phase angle $\varphi$, the pitching amplitude $\theta_m$ is fixed at 45°. (**b**) Lift and power with different pitching amplitude $\theta_m$, the phase angle $\varphi$ is fixed at 90°.

As presented in Figure 19, the aerodynamic performance is closely related to both pitching amplitude and phase angle. The maximum lift appears at $\varphi = 60°$, $\theta_m = 60°$, corresponding to an advanced rotation motion with a relatively higher pitching angle. It should be noted that when the phase angle $\varphi$ is large and exceeds 90° or is less than −90°, negative aerodynamic power appears, which implies the energy harvesting effect. This phenomenon agrees well with the findings of Su et al. [52] on a flapping hydrofoil employed a larger pitching amplitude. Higher power consumption can be found when the pitching motion is isolated ($\theta_m = 0°$) or moves in phase ($\varphi = 0°$) with the sweeping motion. The reason for this might be that a larger wing area facing the flapping direction forms a significant drag force, which induces more power requirement to drive the flapping wing. In other words, flapping with a pitching motion might be a quite saving energy behavior. For the power loading, since the existence of values of power close to 0 leads to non-physical power loading, only the power loading ranges from −20 to 50 are presented in Figure 19c. Obviously, a high aerodynamic efficiency appears at a moderate phase angle around 90°, which has been observed in the kinematics of mostly natural flyers [46,53].

### 3.2.3. $K$ and $C_\theta$

The parameters $K$ and $C_\theta$ determine the kinematic profile of sweeping and pitching respectively. The parametric model in Equations (8) and (9) was designed to mimic the insect flapping kinematics, particularly the pitching movement. From the observation of the inset wing or the flapping robot wing [35,54], it can be found that the pitching motion is similar to the trapezoidal trajectory, characterized by a rapid rotation followed by a slight angle fluctuation, subsequently, maintaining an almost constant pitching angle. And the sweeping motion is very similar to harmonic motion, but there are still some differences.

Figure 20 shows the contours of aerodynamic performance on the $K$-$C_\theta$ plane. $K$ varies from 0.01 to 0.99, and $C_\theta$ varies from 0.01 to 10. Apparently, the impact of $K$ and $C_\theta$ on lift and power appears to be quite different. For the lift, consistent with the findings by Bhat et al. [14], the maximum lift appears at a low $K$ and a high $C_\theta$, corresponding to a sinusoidal sweeping and trapezoidal pitching motion. Furthermore, the influence degree of $K$ and $C_\theta$ on the lift is also different according to the trend of contour lines. Lift increases more rapidly as $C_\theta$ increases, but gradually decreases as $K$ increases, indicating that the insect-inspired pitching motion is beneficial to lift generation. In terms of the aerodynamic power, it seems that $K$ has a stronger influence than $C_\theta$. Lower power consumption can be obtained at a larger $K$, which agrees well with the research of Berman and Wang [34], indicating a triangular sweeping is beneficial to saving energy. The power loading is obviously affected by the shape of pitching motion. A higher power loading can be obtained at a larger $C_\theta$ with a higher $K$. Therefore, it makes sense that this combination, triangular sweeping with trapezoidal pitching motion, appears to be an economical movement for insect flight.

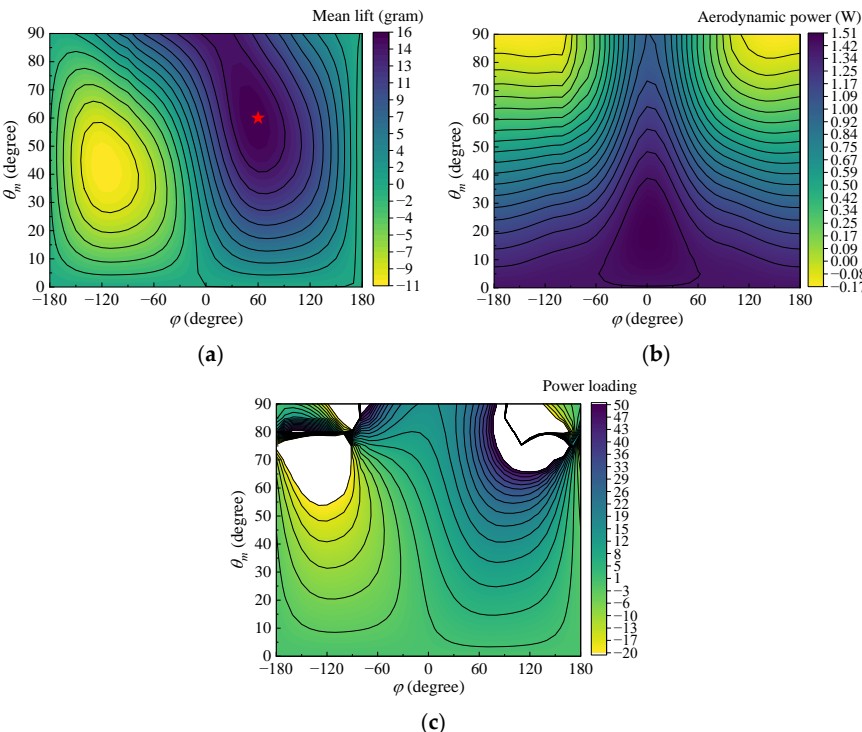

**Figure 19.** Parameter maps of aerodynamic performance as functions of phase angle $\varphi$ and pitching amplitude $\theta_m$. (**a**) Lift with the phase angle and pitching amplitude. The star shape denotes the location of maximum lift. (**b**) Aerodynamic power with the phase angle and pitching amplitude. (**c**) Power loading with the phase angle and pitching amplitude. Note that only the values from $-20$ to 50 are presented in the colored map to isolate the non-physical power loading generated by a near-zero aerodynamic power.

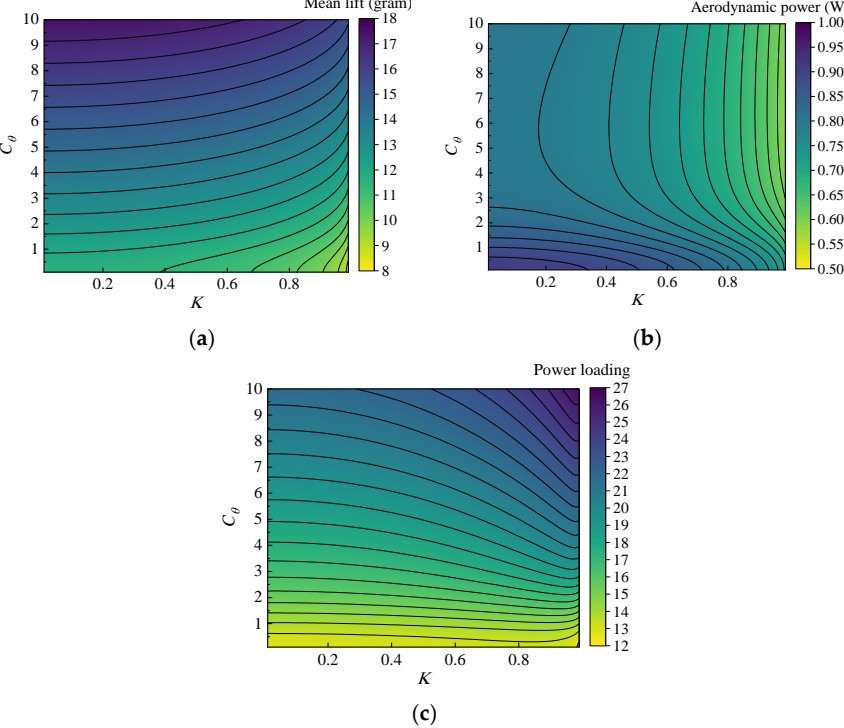

**Figure 20.** Parameter maps of aerodynamic performance as functions of $K$ and $C_\theta$. (**a**) Lift with $K$ and $C_\theta$. (**b**) Aerodynamic power with $K$ and $C_\theta$. (**c**) Power loading with $K$ and $C_\theta$.

As can be seen in Figure 3, the discrepancy in the flapping trajectories for different $K$ and $C_\theta$ are mainly in the angular velocity and acceleration, especially during the stroke reversal, which will directly affect the aerodynamic forces associated with rotational mechanisms [14,50]. Consequently, lift and aerodynamic power components in several typical states are compared and presented in Figure 21. When the pitching profile changes from sinusoidal wave to trapezoidal wave, pitch angular velocity and angular acceleration will increase during the stroke reversal. An extensive increase in rotation and added mass lift can be seen at this stage by referring to Figure 21a,b, resulting in the increase in mean lift force. This sudden increase in angular velocity and angular acceleration also appears in the sweeping motion during the stroke reversal as $K$ increases, which might be responsible for the higher power magnitude at this stage as shown in Figure 21c,d. However, during the translation stage, the angular velocity and angular acceleration of sinusoidal sweeping motion will be larger, leading to more power consumption and, finally, a higher cycle-averaged aerodynamic power.

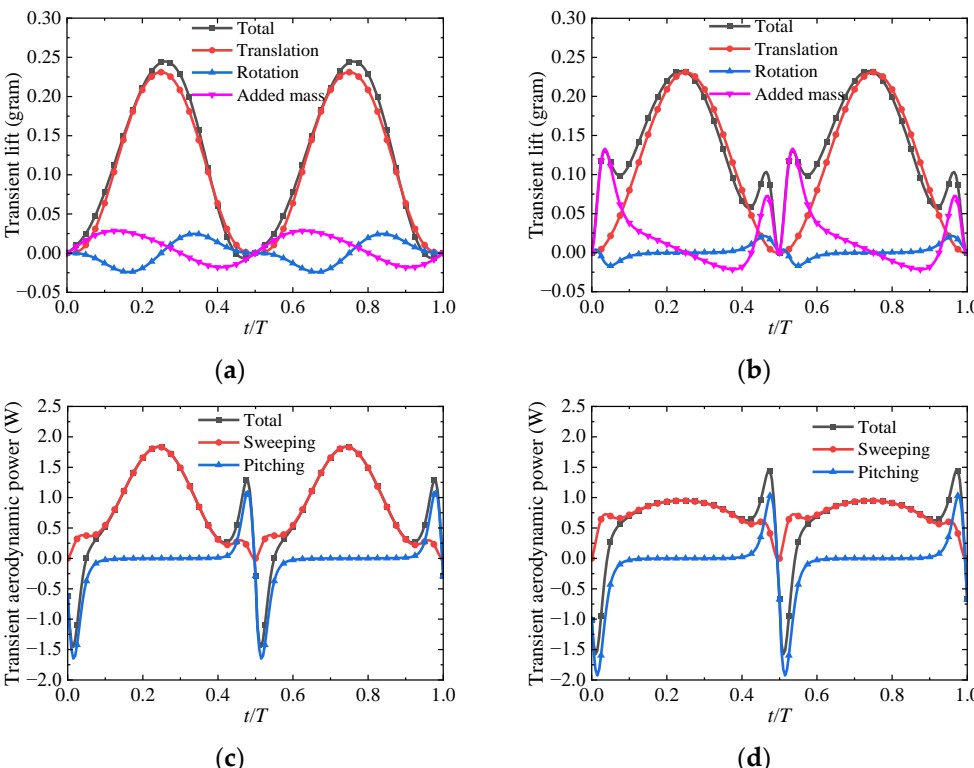

**Figure 21.** Transient aerodynamic performance at different $K$ and $C_\theta$. (**a**) Lift components at $K = 0.01$, $C_\theta = 0.01$. (**b**) Lift components at $K = 0.01$, $C_\theta = 4$. (**c**) Power components at $K = 0.01$, $C_\theta = 4$. (**d**) Power components at $K = 0.9$, $C_\theta = 4$.

## 4. Conclusions

In this study, the sensitivity analysis of wing geometric and kinematic parameters was performed on a hovering flapping wing based on the quasi-steady aerodynamic model. Each parameter's influence on aerodynamic performance was evaluated and compared. Meanwhile, detailed exponential relationships were established between the parameters and the corresponding aerodynamic performance.

For the geometric parameters, sensitivity analysis shows that the wing area $S$ has the greatest influence on lift, and aerodynamic power is most obviously affected by the dimensionless radius of the second moment of area $r_2$. The taper ratio shows less effect on the lift generation than other geometric parameters. All geometric parameters are negatively correlated with the power loading, and increasing area $S$ results in the most gentle decrease in power loading.

For the kinematic parameters, flapping frequency and sweeping amplitude show almost identical effects on the aerodynamic power, but the increase in flapping frequency will lead to faster lift growth and slower drop in power loading. There exists an optimal co-operation of pitching amplitude and phase shift between pitching and sweeping ($\theta_m = 60°$, $\varphi = 60°$) to maximize lift. The pitch motion is beneficial to reduce the power consumption. A higher power loading appears when the phase shift $\varphi$ is close to $90°$.

Lift and power loading are primarily affected by the shape of the pitching motion rather than that of the sweeping motion. The power consumption seems to be dominated by the sweeping motion. The bio-inspired flapping trajectory, characterized by triangular sweeping with trapezoidal pitching motion, is quite an efficient movement.

**Author Contributions:** Conceptualization, B.S.; formal analysis, X.L.; funding acquisition, X.Y.; writing—original draft, X.L.; writing—review & editing, W.Y. and D.X. All authors have read and agreed to the published version of the manuscript.

**Funding:** This study was supported by Shenzhen Science and Technology Program and Research (Grant No. JCYJ 20220530161808018), Guangdong Basic and Applied Basic Research Foundation (Grant No. 208273626031), Basic Research Program of Shenzhen (Grant No. JCYJ 20190806142816524), the Key R&D Program in Shaanxi Province (Grant No. 2023-YBGY-372), the National Natural Science Foundation of China (Grant No. 52175277), the Youth Program of National Natural Science Foundation of China (Grant No. 51905411), the National Key Laboratory of Science and Technology on Aerodynamic Design and Research (Grant No. 61422010301).

**Institutional Review Board Statement:** Not applicable.

**Informed Consent Statement:** Not applicable.

**Data Availability Statement:** Not applicable.

**Conflicts of Interest:** The authors declare no conflict of interest.

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
