# Peer review of "Sensitivity Analysis of Wing Geometric and Kinematic Parameters for the Aerodynamic Performance of Hovering Flapping Wing"

_aerospace, doi:10.3390/aerospace10010074_

Round 1

Reviewer 2 Report

The paper provides a comprehensive analytical approach for understanding effect of geometric and kinematic parameters on flapping wing lift and power requirements. It is a useful study for FWMAV designers. Below are some specific comments for consideration:

1) Eq 17-26 - Please mention reference leading to choice of said coefficients

2) Line 224 - "The cycle averaged inertia power will be zero" - would be good to mention the effect of wing flexibility on inertia power simplification

3) Line 280 - "remarkable flexibility" - suggest rewording to - this study may be limited to application of flapping wings with high rigidity

4) Line 286 -  "the increase of lift appears to be more gentle when the AR is large." - authors can provide more insight on this since Eq.36 suggests linear relationship with AR. Where does non-linearity kick in from the analytical expressions?

5) r2 discussion - the authors could condense this section to keep it focused since it is understood by definition that r2 growth is maximum by moving cmax to the tip.

6) Line 420 - "a non linear relationship can be found between r2 and lift" - please refer to Eq. 36 that can be used to justify this point. 

Round 2

Reviewer 1 Report

Although preferring non-dimensional aerodynamic quantities, I see that the authors have their own viewpoints. I am satisfied with authors' responses and the revised manuscript is sufficiently improved for publication.